# Non-allergic eye rubbing is a major behavioral risk factor for keratoconus

**Katarzyna Jaskiewicz**[1☯], **Magdalena Maleszka-Kurpiel**[2,3☯], **Andrzej Michalski**[3], **Rafal Ploski**[4], **Malgorzata Rydzanicz**[4], **Marzena Gajecka**[1,5]*

1 Institute of Human Genetics, Polish Academy of Sciences, Poznan, Poland, 2 Optegra Eye Health Care Clinic in Poznan, Poznan, Poland, 3 Chair of Ophthalmology and Optometry, Poznan University of Medical Sciences, Poznan, Poland, 4 Department of Medical Genetics, Medical University of Warsaw, Warsaw, Poland, 5 Chair and Department of Genetics and Pharmaceutical Microbiology, Poznan University of Medical Sciences, Poznan, Poland

☯ These authors contributed equally to this work.
* gamar@man.poznan.pl

**Data Availability Statement:** All proceeded clinical/ environmental/behavioral/socioeconomic/ molecular data, as minimal data set underlying the results described in our manuscript, are shared in a

## Abstract

Since the environmental, behavioral, and socioeconomic factors in the etiology of keratoconus (KTCN) remain poorly understood, we characterized them as features influencing KTCN phenotype, and especially affecting the corneal epithelium (CE). In this case-control study, 118 KTCN patients and 73 controls were clinically examined and the Questionnaire covering the aforementioned aspects was completed and then statistically elaborated. Selected KTCN-specific findings were correlated with the outcomes of the RNA-seq assessment of the CE samples. Male sex, eye rubbing, time of using a computer after work, and dust in the working environment, were the substantial KTCN risk factors identified in multivariate analysis, with ORs of 8.66, 7.36, 2.35, and 5.25, respectively. Analyses for genes whose expression in the CE was correlated with the eye rubbing manner showed the enrichment in apoptosis (*TP53*, *BCL2L1*), chaperon-related (*TLN1*, *CTDSP2*, *SRPRA*), unfolded protein response (*NFYA*, *TLN1*, *CTDSP2*, *SRPRA*), cell adhesion (*TGFBI*, *PTPN1*, *PDPK1*), and cellular stress (*TFDP1*, *SRPRA*, *CAPZB*) pathways. Genes whose expression was extrapolated to the allergy status didn't contribute to IgE-related or other inflammatory pathways. Presented findings support the hypothesis of chronic mechanical corneal trauma in KTCN. Eye-rubbing causes CE damage and triggers cellular stress which through its influence on cell apoptosis, migration, and adhesion affects the KTCN phenotype.

## Introduction

Keratoconus (KTCN) is a bilateral corneal disease characterized by progressive thinning and scarring of the cornea, iron deposition, and breaks in Bowman's layer [1]. The usage of state-of-the-art technologies extends this definition to include detected abnormalities in both posterior corneal elevation and corneal thickness distribution, as well as in the corneal epithelium thickness profile [2, 3]. Compound myopic astigmatism is the most prevalent refractive error in the manifest refraction in KTCN patients, followed by hyperopic compound, mixed and

public repository Zenodo (DOI:10.5281/zenodo.
7635600; https://doi.org/10.5281/zenodo.
7635600).

**Funding:** Supported by National Science Centre in Poland (https://www.ncn.gov.pl/en), Grants 2018/31/B/NZ5/03280 (to MG) and 2021/41/B/NZ5/02245 (to MG). Next-generation sequencing was performed thanks to Genomics Core Facility CeNT UW using the NovaSeq 6000 platform financed by the Polish Ministry of Science and Higher Education (decision no. 6817/IA/SP/2018 of 2018-04-10). The funders had no role in study design, data collection and analysis, decision to publish, or preparation of the manuscript.

**Competing interests:** The authors have declared that no competing interests exist.

myopic astigmatism [4]. Visual function support is provided with eyeglasses, contact lenses, or intrastromal corneal ring segments (ICRS) [5]. Progression of KTCN can be modified–slowed down with corneal cross-linking procedure (CXL), which normalizes corneal parameters and improves visual acuity [6]. Penetrating keratoplasty, deep anterior lamellar keratoplasty, and Bowman layer transplantation are treatment options in the advanced stages of the disease [5].

KTCN affects all ethnic groups worldwide and the disease prevalence varies between the described populations. One of the lowest incidence rates (0.0068%) was reported in North Macedonia [7] and the highest (4.79%) in pediatric patients in Saudi Arabia [8]. Based on the performed meta-analysis, the frequency of 1.3/1000 in the general population was estimated [9].

The majority of patients are diagnosed with KTCN at an early age, usually in the second decade of life [10]. In the youngest patients, under 10 years of age, KTCN has been reported with frequent eye rubbing and eye allergies [11, 12].

According to the *Global consensus on keratoconus and ectatic diseases* [2], Down syndrome, connective tissue disorders (Marfan syndrome, Ehlers-Danlos syndrome), Leber congenital amaurosis, floppy eyelid syndrome, Asian and Arabian ethnicity, KTCN diagnosed in family members, ocular allergy, atopy, and the eye rubbing as an exclusive behavior factor, are KTCN risk factors [2]. In addition to the listed genetic syndromes, familial aggregation [13], discordance between twins [14], identified numerous KTCN *loci*, candidate genes and sequence variants [13], and higher KTCN frequency in offspring in communities with high consanguinity rates, confirm the genetic background in KTCN [9, 15]. Therefore, a genetic predisposition together with environmental factors could trigger a biochemical cascade influencing the disease onset [16].

As aspects of the influence of environmental factors on KTCN onset remain a controversial and contentious subject in KTCN research, the aim of this study was to characterize various environmental/behavioral and socioeconomic factors potentially influencing the KTCN phenotype, and especially the corneal surface. We then correlated the clinical/behavioral identified outcomes with the relevant data from analyses of transcriptomic features found in the same assessed patients to further characterize the revealed relationships.

## Materials and methods

### Ophthalmic examination and patients' inclusion and exclusion criteria

This single-center (Optegra Eye Health Care Clinic in Poznan, Poland), prospective, case-control study was conducted from October 2019 to May 2022. The study protocol was approved by the Institutional Review Board at Poznan University of Medical Sciences, Poznan, Poland (resolutions no. 200/22). Subject recruitment, sample collection, and other performed procedures/methods were conducted following the institutional guidelines and regulations, and in accordance with the Declaration of Helsinki. The written informed consent, for study participation and publication without identifying information, before subject recruitment from subjects and/or their legal guardian(s) was obtained. The individual whose photographs are presented in Fig 1 in this manuscript has given written informed consent (as outlined in PLOS consent form) to publish the case details.

Each individual underwent a complete ophthalmological examination, including the assessments of both uncorrected (UDVA) best-corrected distance visual acuity (BDVA), intraocular pressure (IOP), corneal tomography with rotating Scheimpflug camera Pentacam (Oculus Optikgeraete GmbH, Wetzlar, Germany), epithelial thickness mapping MS-39 (CSO, Costruzione Strumenti Oftalmici, Florence, Italy), slit-lamp and dilated funduscopic examination. Additionally, towards screening of the dry eye syndrome symptoms the Ocular Surface Disease

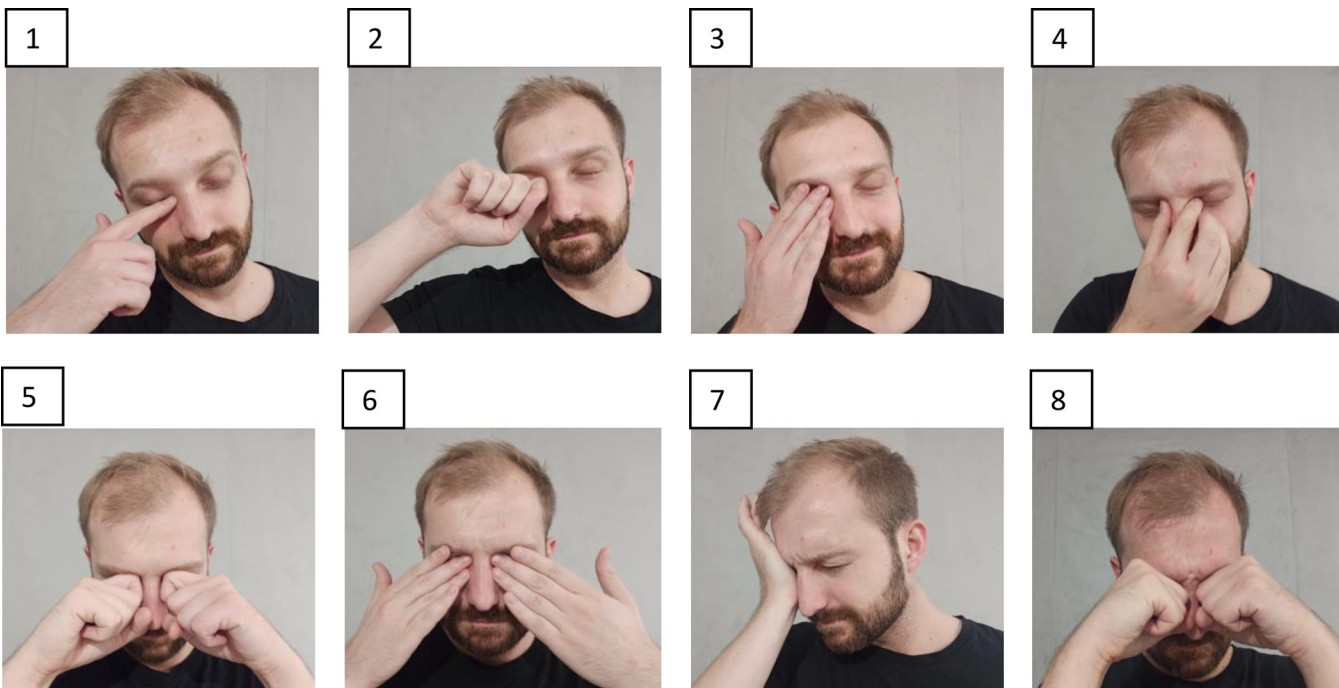

**Fig 1. The identification of the eye rubbing manner in the examined individuals.** The photographs, numbered from 1 to 8, ranked ascending by their severity/potential harmfulness to the eyeball, included in the Questionnaire for the indication of the eye rubbing manner in the examined individuals. In the Questionnaire, the aspect of eye rubbing manner was verified using the multiple question method (Study Questionnaire, question no. 9: '*Please select the photo that most closely resembles the way you rub your eyes*').

Index (OSDI) was evaluated and in suspected individuals, the Tear Breakup Time (TBUT) was checked.

The inclusion criteria for patients with KTCN were the diagnosis of KTCN, the following indices were analyzed: Belin-Ambrósio Enhanced Ectasia Display Total Deviation Value, Ambrósio relational thicknessvalues, pachymetric progression indices, the index of vertical asymmetry, anterior and posterior elevation [17], and corneal epithelial thickness profile [3].

The experienced ophthalmologists subspecialised in cornea and refractive surgery rated all corneal images.

The group of controls consisted of volunteers or regular clinic patients or KTCN family members, who agreed on a long detailed clinical examination followed by a detailed Questionnaire. The inclusion criteria for control individuals included no clinical evidence/symptoms of KTCN.

Exclusion criteria for both studied groups were co-occurring severe ophthalmic condition and genetic diseases/disorders. Additionally, the ongoing ocular surface inflammation was the exclusion criterion.

### Collection of environmental/behavioral/socioeconomic data

A detailed Questionnaire, embracing questions on demographics, behavioral, environmental and socioeconomic aspects, was completed by each participant, in the presence of the researcher. The place of living up to the age of 15 (village or city up to 20,000 residents or city from 20,000 to 100,000 residents or city from 100,000 to 500,000 residents or city with over 500,000 residents); education level (primary or vocational education or high school or university); time of outdoor physical activity up to the age of 15; previous conjunctivitis, dry eye

syndrome, eye trauma, contact lenses use, ocular diseases, allergy (drug/food/pollens/dust/grass), asthma, atopic dermatitis, thyroid gland diseases, genetic diseases, smoking, past pregnancies were recorded.

Detailed questions regarding eye rubbing manner of study individuals embracing frequency of this behavior (occasionally or sometimes or often), dominant hand used, eye preference, preference of a part of hand used to rub, preference of a part of eye to rub, and details about the eye rubbing right after waking up were asked. Additionally, the 8 photographs showing different manners of eye rubbing and ranked ascending by their severity/potential harmfulness to the eyeball (ordinal scale) were presented to the examined person. The order of the photos was determined based on the expertise of ophthalmologists who performed the clinical examination and the researcher processing the surveys.

Also, aspects of professional occupation, presence of dust in working environment (defined as an exposition to filings of metals/building materials/construction materials or excessive dust), excessive UVA exposure, leisure activities (hobbies), reading habits, time spent in front of a screen (computer/tablet/smartphone/electronic reader/television set), hormones intake and sleeping habits (body position, head orientation and hand/arm position) were evaluated.

## Statistical analyses of clinical parameters and environmental/ behavioral/ socioeconomic data

The JASP Software 5 [18] was used in statistical analyses. The normality of the continuous data was assessed by Kołomogorov-Smirnov and Shapiro-Wilk tests. The t-test was applied for continuous variables with a normal distribution. If the normality assumption was not satisfied, the Mann–Whitney test was applied. In the case of nominal data, the Chi-square test was executed.

The odds ratio on univariate logistic regression with 95% confidence intervals were calculated. The multivariate logistic regression was used to create the logistic model of KTCN risk factors, where odds ratios and 95% confidence intervals for variables were recalculated. The confusion matrix (performance diagnostic) of final model was computed, followed by performance metrics including accuracy, area under curve, sensitivity, specificity, and precision. The ROC curve was prepared.

For all performed statistical tests p-values ≤0.05 were considered as statistically significant.

## Integrative analyses and biological interpretation

The integration of the obtained clinical and environmental/behavioral/socioeconomic data with the corresponding transcriptomic experimental data derived from the RNA-seq investigation of the cone (central) region of CE was performed. The transcriptomic assessment of the samples, embracing sample preparation, sequencing, and bioinformatic analyses were performed as previously [19]. Briefly, the CE samples were obtained during CXL [20] and photorefractive keratectomy (PRK) [21] procedures performed in KTCN patients and control individuals undergoing refractive error correction, respectively. The cone (central) region of each CE sample was assessed manually (based on the corneal tomography and epithelial thickness mapping), positioned in the intended orientation on the microscope slide with the use of light microscope and previously made stamps, and separated. Separated CE samples were transferred from the microscope slides to the lysis solution (Norgen Biotek, Thorold, ON, Canada) and total RNA, DNA, and proteins were extracted according to the instructions supplied with the RNA/DNA/Protein Purification Plus Micro Kit (Norgen Biotek). Total RNA libraries were prepared according to a previously established protocol [19], using TruSeq Stranded Total RNA Library Prep Gold (Illumina, San Diego, CA, USA) in accordance with

the manufacturer's protocol. A 100-bp paired-end sequencing run was performed on a Nova-Seq 6000 platform (Illumina). The CE samples were sequenced with an average coverage of 100 million read pairs per sample. Bioinformatic analyses were executed according to a previously established protocol [19].

The relations between expression of particular genes (in TPM, transcript per million) and selected environmental/behavioral/socioeconomic data was executed using Pearson correlation or t-test. The set of dysregulated genes was analyzed using ShinyGO [22], and p-value $\leq 0.05$ and/or FDR $\leq 0.05$ were the cutoffs for pathway enrichment analyses. The Reactome database [23] was used as a pathway term source.

For visualization of the integrative analyses the ggplot2 package [24] in Rstudio (v. 4.1.3) [25] was used.

## Results

### Clinical characteristics of patients and controls

The 118 patients with KTCN (21 females and 97 males) and 73 control individuals (42 females and 31 males) were involved in this study. Among examined control individuals, no signs of KTCN have been disclosed. Clinical characteristics, including UDVA, BDVA, flat keratometric readings (K1), steep keratometric readings (K2), maximum simulated keratometry (Kmax), anterior and posterior elevation of the cornea, thinnest corneal thickness (TCT), thinnest epithelial thickness, axial length (AL), and IOP data of the examined study groups are compiled in Table 1. All presented clinical data differed significantly between the studied groups. No statistical differences in clinical parameters such as K1, K2, Kmax, anterior and posterior elevations, and TCT, have been found in comparison between male and female patients with KTCN, and between adult and adolescent patients with KTCN (S1 and S2 Tables).

The familial occurrence of KTCN concerned 11 patients with KTCN, including two pairs of siblings (brother-brother and sister-sister), two pairs of fathers and their sons, and one trio of an affected father with two daughters with KTCN.

### Behavioral, environmental, and socioeconomic aspects in KTCN

The outcomes of analyzes of selected behavioral, environmental, and socioeconomic aspects evaluated in this study are presented in Tables 2 and 3, together with their statistical significance. All analysed data are compiled in S3 Table, while additional results of comparison of male and female patients with KTCN and adult and adolescent patients with KTCN in aspect of selected behavioral, environmental, and socioeconomic factors are presented in S4 and S5 Tables, respectively. Studied groups have differed in the aspects of age, sex, and place of living, with a predominance of younger males living in villages and minor cities in the KTCN group.

Substantial differences were found in Questionnaire answers concerning the manner of eye rubbing (Table 3). In the multiple questions, the KTCN group more often admitted to the eye rubbing and indicated the fist as part of the hand used for the eye rubbing. Moreover, the patients rubbed most of the eye surface, upper and/or lower eyelid in contrast to the inner and/or outer eye corner rubbed in controls. These behaviors were confirmed by the more frequent selection of the photographs no. 5–8 (as presented in Fig 1) from the remaining photographs included in question no. 9 in the Questionnaire. While we did not found differences in the manner of eye rubbing between adult and adolescent patients with KTCN, there was a significant difference considering sex of patients, that is male patients more frequently used knuckles, a base of hand, all fingers and/or a fist (as presented in photographs no. 5–8) (S4 and S5 Tables). Moreover, in the KTCN group the patients were more frequently left-handed.

**Table 1. Clinical characteristics of the examined patients with KTCN and control individuals.** The statistically significant differences, p ≤0.05 or p <0.001, are indicated.

| | KTCN (n = 118) | | Controls (n = 73) | |
|---|---|---|---|---|
| | x ± SD | Median | x ± SD | Median |
| UDVA OD | 0.46 ± 0.37 | 0.30[‡] | 0.74 ± 0.46 | 0.85 |
| BDVA OD | 0.80 ± 0.35 | 0.90[‡] | 1.12 ± 0.20 | 1.10 |
| UDVA OS | 0.44 ± 0.35 | 0.30[‡] | 0.79 ± 0.46 | 0.95 |
| BDVA OS | 0.78 ± 0.34 | 0.90[‡] | 1.14 ± 0.14 | 1.10 |
| K1 OD [D] | 45.35 ± 5.14 | 43.95[‡] | 42.93 ± 1.34 | 42.90 |
| K2 OD [D] | 48.07 ± 6.11 | 46.10[‡] | 43.86 ± 1.47 | 43.90 |
| Kmax OD [D] | 53.93 ± 9.53 | 51.35[‡] | 44.28 ± 1.53 | 44.30 |
| anterior elevation OD [μm] | 23.55 ± 18.42 | 18.00[‡] | 2.81 ± 1.77 | 2.00 |
| posterior elevation OD [μm] | 50.78 ± 37.95 | 39.50[‡] | 6.60 ± 4.83 | 6.00 |
| TCT OD [μm] | 465.34 ± 56.02 | 471.00[‡] | 542.49 ± 35.51 | 544.00 |
| thinnest epithelial thickness OD [μm] | 43.16 ± 5.45 | 44.00[‡] | 48.59 ± 3.81 | 48.00 |
| K1 OS [D] | 45.45 ± 4.71 | 43.75[‡] | 43.01 ± 1.40 | 43.00 |
| K2 OS [D] | 48.32 ± 5.63 | 46.45[‡] | 43.87 ± 1.59 | 44.10 |
| Kmax OS [D] | 54.54 ± 9.16 | 52.10[‡] | 44.30 ± 1.58 | 44.30 |
| anterior elevation OS [μm] | 26.11 ± 18.40 | 22.50[‡] | 2.82 ± 1.99 | 2.00 |
| posterior elevation OS [μm] | 52.72 ± 32.55 | 49.50[‡] | 6.44 ± 4.15 | 5.50 |
| TCT OS [μm] | 470.64 ± 53.55 | 474.50[‡] | 541.93 ± 35.21 | 542.00 |
| thinnest epithelial thickness OS [μm] | 43.52 ± 5.67 | 44.00[‡] | 49.33 ± 4.08 | 49.00 |
| AL OD [mm] | 24.00 ± 0.89 | 23.95[†] | 23.57 ± 0.98 | 23.40 |
| AL OS [mm] | 23.92 ± 0.88 | 23.85[†] | 23.55 ± 0.92 | 23.40 |
| IOP OD [mmHg] | 13.17 ± 3.18 | 13.00[‡] | 16.46 ± 2.62 | 16.00 |
| IOP OS [mmHg] | 13.13 ± 3.30 | 13.00[‡] | 16.54 ± 2.39 | 16.00 |
| KTCN in a family member | n = 18 (15.25%)[†] | | n = 4 (5.48%) | |
| Any eye disease in a family member | n = 60 (50.85%) | | n = 29 (39.73%) | |

Abbreviations and symbols in the Table: x–average, SD–standard deviation, OD–oculus dexter, OS—oculus sinister, UDVA–uncorrected distance visual acuity, BDVA–best-corrected distance visual acuity, K1 –flat keratometric readings, K2 –steep keratometric readings, Kmax–maximum simulated keratometry, TCT–thinnest corneal thickness, AL—Axial length, IOP–intraocular pressure

[†] indicates statistically significant differences between KTCN patients and controls with p-value ≤0.05

[‡] indicates statistically significant differences between KTCN patients and controls with p-value <0.001

There were no differences in the frequency of reported conjunctivitis, dry eye syndrome, eye trauma, asthma, atopic dermatitis, and allergy to neither food nor pollens/grass/dust, between the studied groups (S3 Table).

Regarding the working environment, despite the lack of difference in the type of professional occupation between studied groups, we found that the presence of dust in the working environment was more frequently indicated by patients with KTCN (Table 2), and especially male patients with KTCN (S4 Table). Moreover, by assessing responses to questions regarding leisure activities, we found that KTCN patients spend more time using a computer after work and a smartphone in the dark (Table 2). There was no difference in the declared hobbies (S3 Table).

Finally, considering the sleeping position including head orientation and hand/arm position, no differences were found comparing the studied groups (S3 Table).

Additional statistics regarding presence of dust in the working environment and eye rubbing, most frequently rubbed eye and right / left-handedness, most frequently rubbed eye and more affected eye, eye rubbing and absolute differences in TCT between left and right eye, eye

**Table 2. Qualitative and quantitative data/results of analyzes of selected behavioral, environmental, and socioeconomic aspects.**

| Variables | | KTCN (n = 118) | Control (n = 73) | p-value |
|---|---|---|---|---|
| Age (years), mean±SD | | 27.3 ± 8.7 | 31.5 ± 10.7 | 0.004 |
| Sex | | | | < 0.001 |
| | Female | 21 (17.78%) | 42 (57.53%) | |
| | Male | 97 (82.20%) | 31 (42.47%) | |
| Level of education | | | | 0.002 |
| | Primary | 14 (11.87%) | 2 (2.74%) | |
| | Vocational education | 11 (9.32%) | 1 (1.37%) | |
| | High school | 51 (43.22%) | 27 (36. 99%) | |
| | University | 42 (35.59%) | 43 (58.90%) | |
| Place of living up to the age of 15 | | | | 0.004 |
| | Village | 43 (36.44%) | 32 (44.44%) | |
| | City up to 20000 residents | 19 (16.10%) | 17 (23.61%) | |
| | City from 20000 to 100000 residents | 7 (5.93%) | 11 (11.11%) | |
| | City from 100000 to 500000 residents | 30 (25.43%) | 3 (4.17%) | |
| | City with over 500000 residents | 19 (16.10%) | 12 (16.67%) | |
| Allergy | | | | 0.128 |
| | Yes | 45 (38.14%) | 20 (27.40%) | |
| | No | 73 (61.86%) | 53 (72.60%) | |
| Food Allergy | | | | 0.223 |
| | Yes | 8 (6.78%) | 2 (2.74%) | |
| | No | 110 (93.22%) | 71 (97.26%) | |
| Pollen/grass/dust Allergy | | | | 0.114 |
| | Yes | 42 (35.59%) | 18 (24.66%) | |
| | No | 76 (64.41%) | 55 (75.34%) | |
| Professional occupation | | | | 0.128 |
| | Student | 27 (23.68%) | 9 (12.50%) | |
| | Non-office worker | 55 (48.25%) | 36 (50.00%) | |
| | Office worker | 32 (28.07%) | 27 (37.50%) | |
| Dust in the working environment | | | | 0.016 |
| | Yes | 34 (28.81%) | 10 (13.70%) | |
| | No | 84 (71.19%) | 63 (86.30%) | |
| Using a computer at work (hours per day) | | 5.7±2.8 | 6.1±2.3 | 0.617 |
| Using a computer after work (hours per day) | | 2.4±1.2 | 1.6±0.9 | < 0.001 |

Note: the respondents did not always answer all the questions, therefore the summation of the answers in individual questions in the table may be incomplete.

rubbing and allergy status, and eye rubbing and time spent using a computer are presented in S6 Table.

## Risk factors for KTCN in univariate analyses

The results of risk analysis of behavioral, environmental, and socioeconomic factors in KTCN, presented as the odds ratio, 95% confidence interval (CI), and p-value are compiled in Table 4. Statistical significance was found for 12 analyzed variables. The male sex (odds ratio 6.26), living in the median size city (100,000–500,000 residents) (OR 6.32), eye rubbing (OR 5.57), frequent rubbing of the left eye (OR 8.91), left handedness (OR 6.39), eye rubbing with fist (OR 4.75), rubbing of both eyes simultaneously with all fingers (photography no. 6; OR 5.73) or fists (photography no. 8; OR 5.52), rubbing of the most of the eye surface with the use of

**Table 3. The data/results of analyzes of the eye rubbing behavior.**

| Variables | | KTCN (n = 118) | Control (n = 73) | p-value |
|---|---|---|---|---|
| Eye rubbing | | | | < 0.001 |
| | Yes | 109 (92.37%) | 50 (68.49%) | |
| | No | 9 (7.62%) | 23 (31.51%) | |
| Frequent eye rubbing | | | | 0.011 |
| | Yes | 10 (8.48%) | 0 (0.00%) | |
| | No | 108 (91.52%) | 73 (100.00%) | |
| Dominant hand | | | | 0.006 |
| | Right | 100 (84.75%) | 71 (97.26%) | |
| | Left | 18 (15.25%) | 2 (2.74%) | |
| More frequently rubbed eye | | | | 0.020 |
| | Both | 79 (72.48%) | 44 (1.67%) | |
| | Right | 14 (12.84%) | 3 (6.25%) | |
| | Left | 16 (14.68%) | 1 (2.08%) | |
| Part of the hand used for rubbing | | | | 0.052 |
| | Fingertips | 37 (41.11%) | 23 (48.94%) | |
| | Base of hand | 2 (2.22%) | 3 (6.38%) | |
| | Knuckles | 29 (32.22%) | 18 (38.30%) | |
| | Fists | 22 (24.45%) | 3 (6.38%) | |
| Eye rubbing with a fist | | | | 0.009 |
| | Yes | 22 (24.44%) | 3 (6.38%) | |
| | No | 68 (75.56%) | 44 (93.62%) | |
| The upper eyelid as the most frequently rubbed part | | | | 0.033 |
| | Yes | 45 (38.14%) | 17 (23.29%) | |
| | No | 73 (61.86%) | 56 (76.71%) | |
| The lower eyelid as the most frequently rubbed part | | | | 0.004 |
| | Yes | 40 (33.90%) | 11 (15.07%) | |
| | No | 78 (66.10%) | 62 (84.93%) | |
| Type of eye rubbing indicated in response to presented photographs | | | | 0.075 |
| | Photography no. 1 | 11 (11.96%) | 14 (28.00%) | |
| | Photography no. 2 | 14 (15.22%) | 11 (22.00%) | |
| | Photography no. 3 | 9 (9.78%) | 4 (8.00%) | |
| | Photography no. 4 | 7 (7.61%) | 7 (14.00%) | |
| | Photography no. 5 | 10 (10.87%) | 4 (8.00%) | |
| | Photography no. 6 | 9 (9.78%) | 2 (4.00%) | |
| | Photography no. 7 | 6 (6.52%) | 2 (4.00%) | |
| | Photography no. 8 | 26 (28.26%) | 6 (12.00%) | |
| Photographs no. 1–4 or 5–8 | | | | 0.002 |
| | Photography 1 or 2 or 3 or 4 | 41 (44.56%) | 36 (72.00%) | |
| | Photography 5 or 6 or 7 or 8 | 51 (55.44%) | 14 (28.00%) | |
| Rubbing the eyes immediately after waking up | | | | 0.191 |
| | Yes | 50 (42.37%) | 24 (32.88%) | |
| | No | 68 (57.63%) | 49 (67.12%) | |

Note: the respondents did not always answer all the questions, therefore the summation of the answers in individual questions in the table may be incomplete.

**Table 4. Selected results of the risk analysis of behavioral, environmental and socioeconomic factors in KTCN.**
For each variable, the odds ratio, 95% confidence interval (CI), and p-value are presented.

| Variable | Odds Ratio | 95% CI | p-value |
|---|---|---|---|
| Sex (*male*) | **6.26** | 3.23–12.13 | **<0.001** |
| Level of education | | | |
| University (*vs* high school) | **0.52** | 0.28–0.97 | **0.041** |
| Place of living up to the age of 15 | | | |
| City with 100,000–500,000 residents (*vs* city with more than 500,000 residents) | **6.32** | 1.57–25.34 | **0.009** |
| Allergy (*yes*) | 1.63 | 0.87–3.08 | 0.130 |
| Food allergy (*yes*) | 2.58 | 0.53–12.51 | 0.239 |
| Pollen/grass/dust allergy (*yes*) | 1.69 | 0.88–3.24 | 0.115 |
| Asthma (*yes*) | 1.60 | 0.48–5.29 | 0.444 |
| Smoking (*yes*) | 0.86 | 0.421–1.77 | 0.686 |
| Eye rubbing (*yes*) | **5.57** | 2.41–12.91 | **<0.001** |
| Dominant hand (*left*) | **6.39** | 1.44–28.41 | **0.015** |
| More frequently rubbed eye | | | |
| Left (*vs both*) | **8.91** | 1.14–69.47 | **0.037** |
| Eye rubbing with fists (*yes*) | **4.75** | 1.34–16.80 | **0.016** |
| Rubbing of the upper eyelid (*yes*) | 1.39 | 0.70–2.79 | 0.349 |
| Rubbing of the lower eyelid (*yes*) | 2.08 | 0.96–4.50 | 0.064 |
| Type of eye rubbing indicated in response to presented photographs | | | |
| Photography no.6 (*vs* No.1) | **5.73** | 1.02–32.10 | **0.047** |
| Photography no.8 (*vs* No.1) | **5.52** | 1.68–18.10 | **0.005** |
| Photographs no. 1–4 or 5–8 (*vs* No. 5–8) | **3.20** | 1.52–6.72 | **0.002** |
| Working environment with dust (*yes*) | **2.55** | 1.17–5.55 | **0.018** |
| Time of using a computer at work (*hours per day*) | 0.95 | 0.82–1.09 | 0.438 |
| Time of using a computer after work (*hours per day*) | **1.91** | 1.31–2.77 | **<0.001** |
| Using a smartphone (*yes*) | 2.21 | 0.79–6.22 | 0.133 |
| Time of using a smartphone during the day (*hours per day*) | 1.20 | 0.93–1.54 | 0.154 |
| Using a smartphone during the night (*yes*) | 0.88 | 0.47–1.62 | 0.671 |
| Time of using a smartphone during the night (*hours per day*) | 1.51 | 0.85–2.68 | 0.157 |

greater force (rubbing with knuckles/base of hand /all fingers /fists, photographs no. 5–8; OR 3.20), presence of dust in the working environment (OR 2.55), and longer time of using a computer after work (OR 1.91) were factors recognized as significantly increasing the KTCN risk. Higher education (university; OR 0.52) was the only variable reducing the risk of KTCN.

In addition, in Table 4 the non-significant but previously discussed factors (such as allergy or asthma) are presented.

## Multivariate analysis and the resulting KTCN risk factor model

Male sex, eye rubbing, time of using a computer after work, and dust in the working environment, were the substantial KTCN risk factors identified in multivariate analysis, with OR

**Table 5. Substantial KTCN risk factors, odds ratios, the confusion matrix, and performance metrics in a multivariate analysis.**

**A. Odds ratios**

| Variable | Odds Ratio | 95% CI | p-value |
|---|---|---|---|
| Sex (*male*) | 8.66 | 3.16–23.79 | <0.001 |
| Eye rubbing (*yes*) | 7.36 | 2.02–26.89 | 0.003 |
| Time of using a computer after work (*hours per day*) | 2.35 | 1.42–3.88 | <0.001 |
| Dust in the working environment (*yes*) | 5.25 | 1.23–22.33 | 0.025 |

**B. Confusion matrix**

| | **Predicted** | | |
|---|---|---|---|
| **Observed** | **Control** | **KTCN** | **% Correct** |
| Control | 39 | 14 | 73.585 |
| KTCN | 11 | 67 | 85.897 |
| Overall % Correct | | | 80.916 |

**C. Performance metrics**

| | Value |
|---|---|
| Accuracy | 0.809 |
| AUC | 0.882 |
| Sensitivity | 0.859 |
| Specificity | 0.736 |
| Precision | 0.827 |

values of 8.66, 7.36, 2.35, and 5.25, respectively. The parameters of the recognized model, including the confusion matrix and performance metrics (accuracy, Area Under the Curve (AUC), sensitivity, specificity, and precision) are presented in Table 5A–5C, whereas the Receiver Operating Characteristic (ROC) curve for these variables in the logistic model, is shown in Fig 2.

## Effect of eye rubbing on corneal surface transcriptome and biological interpretation

The results of the integration of transcriptomic data of the cone (central) region of the corneal epithelium (CE) of 23 KTCN patients and five controls and chosen environmental and behavioral factors are presented in Figs 3–5 and Table 6.

The pathway enrichment analysis for genes whose expression in the cone region of CE was found to be correlated with the eye rubbing manner (recognized by selecting one of the photographs presented in Fig 1 by study individuals) showed the enrichment in apoptosis, chaperon-related, and unfolded protein response pathways (Fig 3). Particularly, the expression of *TGFBI* (R = 0.79, p = 0.0039), *BCL2L1* (R = 0.65, p = 0.31), *CAPZB* (R = 0.73, p = 0.011), *CTDSP2* (R = 0.74, p = 0.0086), *PTPN1* (R = 0.77, p = 0.0057), *PDPK1* (R = 0.8, p = 0.0029), *RAPGEF3* (R = 0.79, p = 0.0041), *TFDP1* (R = 0.68, p = 0.022), *SRPRA* (R = 0.72, p = 0.013), *TLN1* (R = 0.62, p = 0.042), and *TP53* (R = 0.78, p = 0.0042) genes, was correlated with severity/potential harmfulness of the eye rubbing to the eyeball (Fig 4).

The expression of *TRIM8* (R = 0.73, p = 0.0006), *JUNB* (R = 0.64, p = 0.0043), and *LTBP3* (R = 0.66, p = 0.0028) genes were found to be correlated with time spent using a computer after work, as presented in Fig 5.

Without taking into account the environmental/behavioral aspects in the correlation analysis, the expression of all the genes mentioned above did not differentiate the studied groups. (S1 Fig).

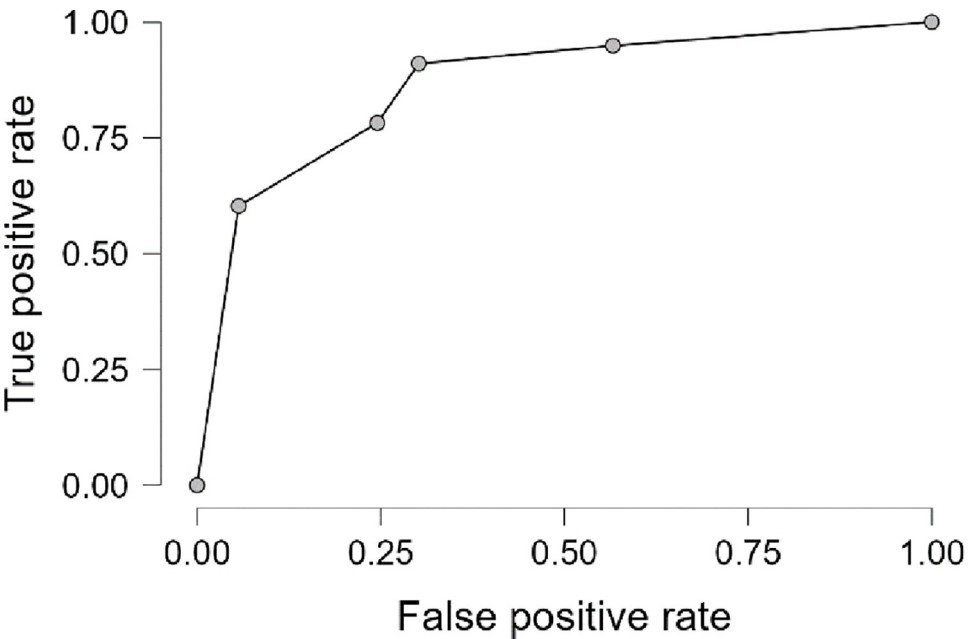

**Fig 2. ROC curve of multivariate model of KTCN risk factors.** ROC curve for the multivariate logistic regression analysis illustrating the performance of KTCN risk factors model. Model statistic values are presented in Table 5.

## Influence of allergy status on transcriptomic features and biological interpretation

Evaluating the allergic / non-allergic factors contributing to the phenotype we found that the genes whose expression was extrapolated to the allergy status didn't contribute to IgE-related pathways, such as 'allergen dependent IgE bound FCERI aggregation'. No allergy-related inflammation of any kind has occurred (e.g., 'interleukin-4 and interleukin-13 signaling'), as presented in S2 Fig, and S7 Table.

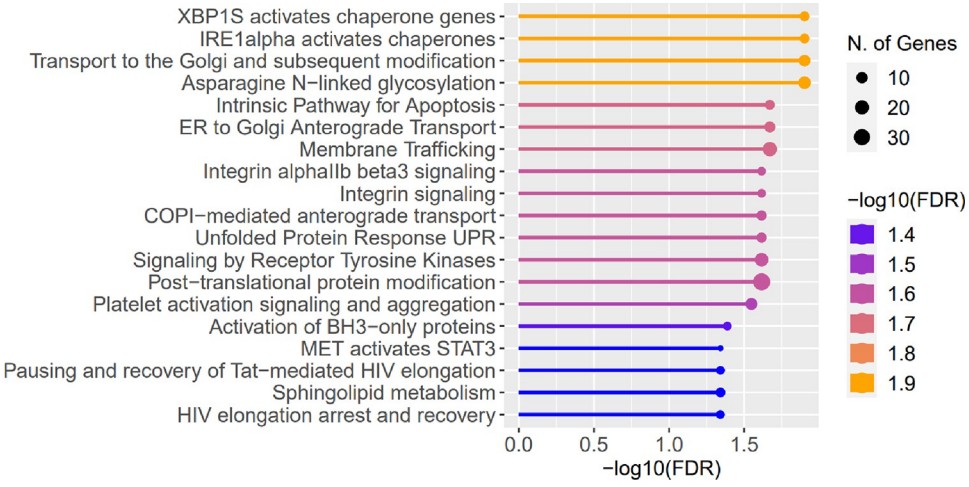

**Fig 3. Results of pathway enrichment analysis.** Results of pathway enrichment analysis for genes which expression in corneal epithelium was found to be correlated with the eye rubbing manner. The type of eye rubbing was indicated in response to presented photographs, included in Fig 1. Transcriptome data was obtained for males and females from both study groups. The Reactome database was chosen as the source of the pathways.

**Table 6. The results of the pathway enrichment analysis.** The results of the pathway enrichment analysis for genes whose expression in the cone region of corneal epithelium was found to be correlated with the eye rubbing manner.

| Pathway | Fold Enrichment | Enrichment FDR | nGenes | Pathway Genes | Genes |
|---|---|---|---|---|---|
| Asparagine N-linked glycosylation | 3.5712 | 0.0126 | 14 | 305 | *CTSA, CAPZB, ARFGAP1, NAPA, ST3GAL4, GORASP1, TBC1D20, TMEM115, PRKCSH, B4GALT3, DCTN5, LMAN2, MAN1B1, DYNLL2* |
| Transport to the Golgi and subsequent modification | 4.6261 | 0.0126 | 11 | 185 | *CAPZB, ARFGAP1, NAPA, ST3GAL4, GORASP1, TBC1D20, TMEM115, B4GALT3, DCTN5, LMAN2, DYNLL2* |
| XBP1S activates chaperone genes | 9.7253 | 0.0126 | 6 | 48 | *ARFGAP1, WFS1, KLHDC3, TLN1, CTDSP2, SRPRA* |
| IRE1alpha activates chaperones | 8.8078 | 0.0126 | 6 | 53 | *ARFGAP1, WFS1, KLHDC3, TLN1, CTDSP2, SRPRA* |
| Membrane Trafficking | 2.4349 | 0.0214 | 21 | 671 | *DVL2, AP2B1, CAPZB, PACSIN2, AP1B1, ARFGAP1, STX4, STX10, ANKRD27, NAPA, GORASP1, TBC1D20, TMEM115, KIF1C, VPS4A, DCTN5, YWHAB, LMAN2, ALS2CL, AP1G2, DYNLL2* |
| Intrinsic Pathway for Apoptosis | 7.6527 | 0.0214 | 6 | 61 | *TP53, YWHAB, STAT3, BCL2L1, TFDP1, DYNLL2* |
| ER to Golgi Anterograde Transport | 4.5469 | 0.0214 | 9 | 154 | *CAPZB, ARFGAP1, NAPA, GORASP1, TBC1D20, TMEM115, DCTN5, LMAN2, DYNLL2* |
| Post-translational protein modification | 1.8475 | 0.0242 | 36 | 1516 | *SCMH1, CTSA, OTUD5, CAPZB, PIGS, JOSD1, ARFGAP1, AXIN1, NAPA, TNKS2, DPH1, WFS1, ST3GAL4, GORASP1, TBC1D20, TMEM115, PRKCSH, DDA1, UBE2M, TRIM28, CKAP4, TP53, USP21, PEX10, B4GALT3, UBE2Z, DCTN5, LMAN2, MAN1B1, BTBD6, POMK, WDR5, H4-16, GPAA1, RAET1G, DYNLL2* |
| Unfolded Protein Response UPR | 5.3394 | 0.0242 | 7 | 102 | *NFYA, ARFGAP1, WFS1, KLHDC3, TLN1, CTDSP2, SRPRA* |
| COPI-mediated anterograde transport | 5.3922 | 0.0242 | 7 | 101 | *CAPZB, ARFGAP1, NAPA, GORASP1, TMEM115, DCTN5, DYNLL2* |
| Signaling by Receptor Tyrosine Kinases | 2.5233 | 0.0242 | 18 | 555 | *AP2B1, ATP6AP1, COL5A3, PXN, POLR2C, MET, PPP2R5D, YAP1, DUSP6, PDPK1, GRB7, POLR2D, MAPKAPK2, YWHAB, STAT3, NELFB, PTPN1, COL5A2* |
| Integrin alphaIIb beta3 signaling | 11.5262 | 0.0242 | 4 | 27 | *RAPGEF3, TLN1, PDPK1, PTPN1* |
| Integrin signaling | 11.5262 | 0.0242 | 4 | 27 | *RAPGEF3, TLN1, PDPK1, PTPN1* |
| Platelet activation signaling and aggregation | 3.1648 | 0.0283 | 12 | 295 | *STXBP2, GNB1, RAPGEF3, GNA11, STX4, GNAI2, TLN1, PDPK1, CHID1, PTPN1, PSAP, ACTN4* |
| Activation of BH3-only proteins | 9.7253 | 0.0409 | 4 | 32 | *TP53, YWHAB, TFDP1, DYNLL2* |
| Pausing and recovery of Tat-mediated HIV elongation | 9.1532 | 0.0455 | 4 | 34 | *POLR2C, CDK9, POLR2D, NELFB* |
| Tat-mediated HIV elongation arrest and recovery | 9.1532 | 0.0455 | 4 | 34 | *POLR2C, CDK9, POLR2D, NELFB* |
| Sphingolipid metabolism | 5.2451 | 0.0455 | 6 | 89 | *CTSA, KDSR, CSNK1G2, SMPD1, CERS6, PSAP* |
| MET activates STAT3 | 38.9010 | 0.0455 | 2 | 4 | *MET, STAT3* |
| HIV elongation arrest and recovery | 8.6447 | 0.0456 | 4 | 36 | *POLR2C, CDK9, POLR2D, NELFB* |
| Pausing and recovery of HIV elongation | 8.6447 | 0.0456 | 4 | 36 | *POLR2C, CDK9, POLR2D, NELFB* |
| MHC class II antigen presentation | 4.3569 | 0.0456 | 7 | 125 | *AP2B1, CTSA, CAPZB, AP1B1, ACTR1B, DCTN5, DYNLL2* |

## Discussion

The genetic predisposition to KTCN, comprising the presence of sequence variation in genes related to KTCN [26, 27] and KTCN-specific transcriptomic alterations [19, 28] along with environmental [9, 29], behavioral [9, 30, 31], and other unknown factors, most likely cause the onset of KTCN.

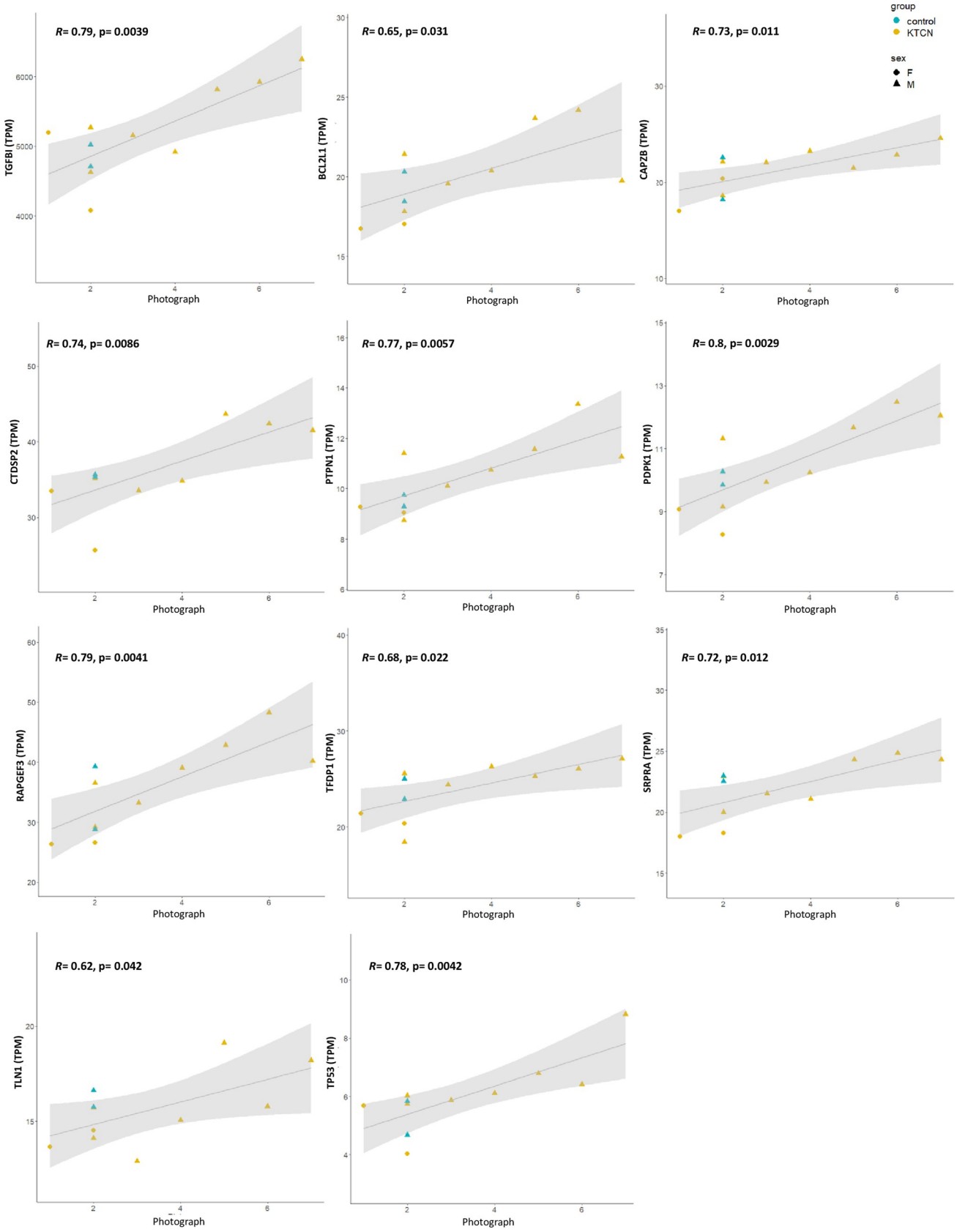

**Fig 4. Results of Pearson's correlation.** Pearson's correlation results between the expression of selected genes, *TGFBI*, *BCL2L1*, *CAPZB*, *CTDSP2*, *PTPN1*, *PDPK1*, *RAPGEF3*, *TFDP1*, *SRPRA*, *TLN1*, and *TP53* (in TPM), in the cone (central) region of corneal epithelium and severity/potential harmfulness of the eye rubbing to the eyeball recognized by selecting one of the photographs presented in Fig 1 by study individuals. Values of regression and statistical significance are presented in plots.

Previously the strong relation between eye rubbing and KTCN was reported in the case reports [32], case series [33], multivariate analyses [30], and meta-analysis [9]. Here we verified this aspect and further characterized the eye rubbing behavior practiced by patients with KTCN. Patients used their fists more frequently to rub the eye, and they also rubbed most of the eye surface, i.e., the upper and/or lower eyelid, as opposed to the inner and/or outer corner of the eye. Moreover, this disclosed manner involved the use of greater force when patients with KTCN rubbed their eyes. Previously, the mechanical forces exerted on the eyelids by KTCN patients by rubbing the eyes were reported to be different, depending on which part of the hand was used [34].

Comparing the corneal thickness and other clinical parameters between less and more frequently rubbed eyes, we didn't find the substantial differences. Although we've revealed relation between eye rubbing and absolute differences in TCT (TCT diff) between right and left eye, the specific manner of eye rubbing (e.g., eye rubbing with fist) or its severity did not show correlation with this variable. Since central corneal thickness is one of the most heritable human characteristics [35], we concluded that the effect of eye rubbing on the KTCN phenotype was rather due to the generation of cellular stress in the CE.

Previously, *in vitro* study showed the alteration of cytoskeleton and migratory behavior of corneal epithelial cells exposed to shear stress [36]. In line with these findings, here the increased expression of *TFDP1*, *SRPRA* and *CAPZB*, being elements of cellular responses to stress, and the increased expression of *TLN1*, *RABGEF3*, *PDPK1*, and *PTPN1*, being elements of extracellular matrix (ECM) affecting cell adhesion and migration were revealed, as the *in vivo* evidence of influence of severe eye rubbing on CE. Importantly, these results were consistent for the patients of both sexes, while previously reported transcriptomic data [37] and epidemiological [38] observations were sex-specific.

Moreover, we revealed the important eye rubbing-related changes in expression of genes contributing to apoptosis (*BCL2L1*, *TP53*, *BAX*), chaperone activation (*TLN1*, *CTDSP2*, *SRPRA*), and unfolded protein response (*NFYA*, *TLN1*, *CTDSP2*, *SRPRA*). Previously, among the listed genes, high levels of protein p53 were found in unaffected CE of various vertebrate

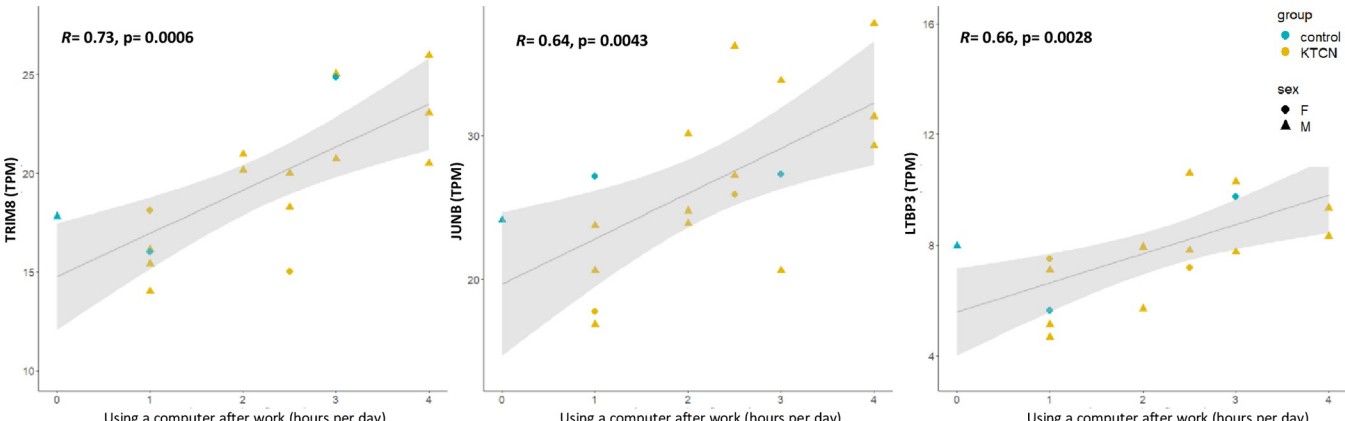

**Fig 5. Results of Pearson's correlation.** Pearson's correlation results between the expression of selected genes, *TRIM8*, *JUNB*, and *LTBP3* (in TPM), in the cone (central) region of corneal epithelium and time spent using a computer after work. Values of regression and statistical significance are presented in plots.

species [39], and more importantly in the interpretation of our outcomes, the experimental study showed the increased cytoplasmic expression of p53 after the cell stress stimulation resulting from ultraviolet irradiation [40].

Notably, in a relation to severity of eye rubbing a dysregulated expression of elements of the TGFB signaling pathway was identified, previously reported as the KTCN-specific feature [19] and discussed as influencing the activation of corneal cells to produce various types of collagen and other ECM components [41]. Upregulation of *TGFBI* revealed here was found to be corelated with more severe eye rubbing, whereas increased expression of *JUNB* and *LTBP3* were strongly related to the hours spent in front of the computer after work. The longer time of using a computer after work was revealed as an independent risk factor for KTCN. However, patients with KTCN were younger and definitely more addictive to electronics and that should be considered in the data interpretation.

Moreover, the presence of dust in the working environment increased the risk of both the eye rubbing and KTCN itself. Interestingly, this factor may have similar influence on corneal surface as the living in median size cities, in which exposure to air pollution, particularly to particles with size less than 2.5 μm, was reported to be higher as an effect of urbanization degree and household heating systems [42, 43].

Here, the allergy has not been disclosed as a risk factor for KTCN. Regarding the aspect of allergy and atopy in patients with KTCN, it has been previously discussed as more prevalent in KTCN than in general population [30, 44, 45]. However, in some reports using multivariate analyses and meta-analyses, the atopy and allergic eye diseases were not significantly associated with KTCN [46, 47] and in many developing countries the constant increase in occurrence of allergic and atopic diseases has been reported [48, 49]. For example, the incidence of at least one allergic condition changed in years 2001–2005 from 18.9% to 24.2% in patients registered in England database (QRESEARCH 2,958,366 patients) [50]. In the analysis of transcriptome in the full-thickness corneas no significance for the allergy status was found, no differentially expressed genes comparing the KTCN patients with allergy vs. without allergy were indicated [51]. Moreover, no significant differences in the ocular surface immune cell subset proportions and tear fluid soluble factor levels were observed between KTCN eyes of different grades [52]. Also in that study, the influence of ocular allergy, history of systemic allergy and eye rubbing on the immune cell profile was investigated, and no differences were found [52]. In contrast, the significantly higher tear fluid levels of IL-1α, IL-9, IL-10, IL-13, TNFα, sVCAM, sTNFRII and IgE were observed in KTCN patients with history of systemic allergy and significantly higher levels of cytokines (IL-2, IL-12/23p40, IL-12p70, IL-13, IL-17A, IFNα), chemokines (MIG/CXCL9; ITAC/CXCL11), and growth factors (TGFβ1, EPO, VEGF) were revealed in KTCN patients with a history of eye rubbing [52]. In our KTCN patients the allergy was found to be not related to the severity of eye rubbing. Importantly, allergy status has not affected CE transcriptomic outcomes obtained for the same evaluated patients. Also, other inflammatory conditions, reported in the past in medical history, conjunctivitis, both bacterial and viral and autoimmune diseases, were not found to be more frequently observed in patients with KTCN. We conclude here that the eye rubbing behavior is not a consequence of the allergic condition.

We observed that patients with KTCN were more frequently left-handed comparing to control individuals being right-handed in general. Although we did not find a relation between the most frequently rubbed eye and right/left-handedness, the relation between the most frequently rubbed eye and the more affected eye, in particular between the left eye and the preference of left-eye rubbing, was revealed.

It has been reported that patients with KTCN more frequently than controls manifested dry eye syndrome [53]. Here the relation of KTCN with the dry eye syndrome, verified by TBUT and/or OSDI (data not shown), was not confirmed.

As outdoor activity is a verified factor influencing other multifactorial eye diseases, the myopia and high myopia phenotypes [54], and it has not been investigated in KTCN research, we also assessed this aspect in patients with KTCN as a risk or protective environmental factor and found it as not differentiating between the groups of patients and controls which we evaluated.

Also, patients with KTCN less frequently read paper books and used the electronic readers, probably due to the joint effect of a generational change as well as a problem with visual acuity that cannot be adjusted through glasses or contact lenses.

In this paper, we presented the level of higher education as the only variable reducing the risk of KTCN (OR = 0.52, univariate analysis), but this result should be treated with caution as it may be a consequence of selection bias.

Other, previously discussed factors as sleeping position and creating additional pressure on the eyeball [30] or smoking [31] weren't confirmed in our study. Moreover, the effect of pregnancy on KTCN progression could not be verified due to a small number of female patients in KTCN group. In this study more men than women were ascertained in KTCN group, which is in line with the predominance of male patients among 110 individuals undergoing the CXL procedure in 2020–2022 in Optegra Clinic (19 female and 91 male patients, unpublished data, Maleszka-Kurpiel, MD, PhD, personal communication) and with the previous studies [55, 56].

As this is a single center patient recruitment study, we interpreted our results with caution. Questions asked, based on the applied Questionnaire, were somehow biased, for example in recalling of habits/behaviors and quantifying their degree. Therefore, in the case of the behavioral aspects, we used the multiple question method to minimize such biases. In addition, the study groups were not matched in terms of age and sex, as the aim of the study was also to analyze the impact of these characteristics. We did not identify clinical features corelated with sex or age, however, the sex and age could potentially influence study results regarding the life habits. The absence of differences in clinical parameters between adult and adolescent patients indicates a rapid progression in the latter [57–59], pointing to the importance of increasing patient awareness [60] as well as early diagnosis due to substantial risk of progression. Generally, we focused on the regular patients, who best represent the overall patient population of the clinic. Hence in our control group, the KTCN family members occur, but differences in their life habits in comparison to KTCN-affected family members could be a valuable practical and clinical information. Importantly, the study groups do not differ in the aspect of contact lens wear, which could have influence the issue of frequency of the eye rubbing. Moreover, the absence of differences in clinical parameters between adult and adolescent patients, and male and female patients, in contrast to previous reports [38, 61], enabled a superior appreciation of both, behavioral and molecular findings. Regarding other limitations, obtaining the CE samples from the control individuals towards transcriptomic assessment was especially challenging, as this type of material could be collected during PRK procedure only, which nowadays is rarely performed and replaced with the newer techniques of vision correction, in which the CE is not removed.

So far, the impact of shear stress on function/mechanics of epithelial cells in single-layer model [36] and changes in mechanical forces exerted on the eye depending on eye-rubbing manner [34] have been reported. However, currently the effects of repetitive eye rubbing on corneal biomechanics have not been confirmed in a more complex model such as the ex-vivo porcine eyes [62]. Therefore, the influence of the eye rubbing, confirmed in the corneal epithelium cultured cells under flow conditions [36] should be further investigated in the studies on the biological mechanisms of this behavior on the physiology of the cornea, involving the 3D experimental models.

Overall, environmental / behavioral factors were shown to have a significant influence on the corresponding KTCN CE transcriptomic findings in the patients with KTCN. More importantly, the evaluated transcriptomic differences found between KTCN and control samples were only revealed after taking environmental / behavioral factors into account.

In conclusion, the male sex, eye rubbing, time of using a computer after work, and the dust in the working environment were found or verified as the most significant risk factors for KTCN and the effect of selected environmental/behavioral features on ocular surface in KTCN was revealed. Presented transcriptomic findings, integrated with environmental/ behavioral aspects, support the hypothesis of chronic mechanical corneal trauma affecting the pathogenesis of KTCN. This trauma in the form of eye rubbing-caused damage of the CE, leaving the allergy as a redundant condition, contributes to the cellular stress, which through influence on cell apoptosis, migration, and adhesion affects the KTCN phenotype.

## Supporting information

**S1 Fig. The boxplots of selected genes' expression.** The boxplots presenting the expression of selected genes, *TGFBI*, *BCL2L1*, *CAPZB*, *CTDSP2*, *PTPN1*, *PDPK1*, *RAPGEF3*, *TFDP1*, *SRPRA*, *TLN1*, *TP53*, *TRIM8*, *JUNB*, and *LTBP3* (in TPM, Transcripts per Million), in the cone (central) region of corneal epithelium in the studied groups of patients with KTCN and controls.
(TIF)

**S2 Fig. Results of pathway enrichment analysis.** Results of pathway enrichment analysis for genes whose expression in corneal epithelium was found to be correlated with the allergy status. Transcriptome data was obtained for males and females from both study groups. The Reactome database was chosen as the source of the pathways.
(TIF)

**S1 Table. Results of clinical comparison of male and female patients with KTCN.**
(DOCX)

**S2 Table. Results of clinical comparison of adult and adolescent patients with KTCN.**
(DOCX)

**S3 Table. Complete information on the qualitative and quantitative data/results of analyzes of behavioral, environmental, and socioeconomic aspects.**
(DOCX)

**S4 Table. Results of comparison of male and female patients with KTCN in aspect of selected behavioral, environmental, and socioeconomic factors.**
(DOCX)

**S5 Table. Results of comparison of adult and adolescent patients with KTCN in aspect of selected behavioral, environmental, and socioeconomic factors.**
(DOCX)

**S6 Table. Additional statistical analyses.**
(DOCX)

**S7 Table. Results of pathway enrichment analysis for genes whose expression in corneal epithelium was found to be related to the allergy status.**
(DOCX)

**S8 Table. STROBE statement—checklist.**
(DOC)

## Acknowledgments

We are grateful to the director and the team of the Optegra Eye Health Care Clinic in Poznan, Poland, for enabling and assisting in patient recruitment and clinical evaluation.

## Author Contributions

**Conceptualization:** Katarzyna Jaskiewicz, Magdalena Maleszka-Kurpiel, Marzena Gajecka.

**Data curation:** Katarzyna Jaskiewicz.

**Formal analysis:** Katarzyna Jaskiewicz.

**Funding acquisition:** Marzena Gajecka.

**Investigation:** Katarzyna Jaskiewicz, Magdalena Maleszka-Kurpiel, Andrzej Michalski, Malgorzata Rydzanicz.

**Methodology:** Katarzyna Jaskiewicz, Magdalena Maleszka-Kurpiel, Marzena Gajecka.

**Project administration:** Marzena Gajecka.

**Resources:** Magdalena Maleszka-Kurpiel, Andrzej Michalski.

**Supervision:** Rafal Ploski, Marzena Gajecka.

**Visualization:** Katarzyna Jaskiewicz.

**Writing – original draft:** Katarzyna Jaskiewicz, Magdalena Maleszka-Kurpiel, Andrzej Michalski, Malgorzata Rydzanicz.

**Writing – review & editing:** Rafal Ploski, Marzena Gajecka.

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
