## [Decision Letter · Decision Letter 0]

3 Feb 2023

PONE-D-22-34939Non-allergic eye rubbing is a major behavioral risk factor for keratoconusPLOS ONE

Dear Dr. Gajecka,

Thank you for submitting your manuscript to PLOS ONE. After careful consideration, we feel that it has merit but does not fully meet PLOS ONE’s publication criteria as it currently stands. Therefore, we invite you to submit a revised version of the manuscript that addresses the points raised during the review process.

We look forward to receiving your revised manuscript.

Kind regards,

Georgios Labiris, MD, PhD

Academic Editor

PLOS ONE

Journal Requirements:

"Supported by National Science Centre in Poland (https://www.ncn.gov.pl/en), Grants 2018/31/B/NZ5/03280 (to MG) and 2021/41/B/NZ5/02245 (to MG)."

"Next-generation sequencing was performed thanks to Genomics Core Facility CeNT UW using NovaSeq 6000 platform financed by the Polish Ministry of Science and Higher Education (decision no. 6817/IA/SP/2018 of 2018-04-10)."

"Supported by National Science Centre in Poland (https://www.ncn.gov.pl/en), Grants 2018/31/B/NZ5/03280 (to MG) and 2021/41/B/NZ5/02245 (to MG)."

7. We note that Figure 1 includes an image of a participant in the study. 

As per the PLOS ONE policy (http://journals.plos.org/plosone/s/submission-guidelines#loc-human-subjects-research) on papers that include identifying, or potentially identifying, information, the individual(s) or parent(s)/guardian(s) must be informed of the terms of the PLOS open-access (CC-BY) license and provide specific permission for publication of these details under the terms of this license. Please download the Consent Form for Publication in a PLOS Journal (http://journals.plos.org/plosone/s/file?id=8ce6/plos-consent-form-english.pdf). The signed consent form should not be submitted with the manuscript, but should be securely filed in the individual's case notes. 

Please amend the methods section and ethics statement of the manuscript to explicitly state that the patient/participant has provided consent for publication: “The individual in this manuscript has given written informed consent (as outlined in PLOS consent form) to publish these case details”. 

8. Please upload a new copy of Figure 3 to 4 and Supporting Figure S1 as the detail is not clear. Please follow the link for more information:

https://blogs.plos.org/plos/2019/06/looking-good-tips-for-creating-your-plos-figures-graphics/

https://blogs.plos.org/plos/2019/06/looking-good-tips-for-creating-your-plos-figures-graphics/

**Additional Editor Comments:**

Both reviewers have indicated that the paper has potential after some corrections. Please read their comments below and address their suggestions.

Reviewers' comments:

Reviewer's Responses to Questions

**Comments to the Author**

1. Is the manuscript technically sound, and do the data support the conclusions?

Reviewer #1: Yes

Reviewer #2: Yes

2. Has the statistical analysis been performed appropriately and rigorously? 

Reviewer #1: Yes

Reviewer #2: Yes

3. Have the authors made all data underlying the findings in their manuscript fully available?

Reviewer #1: Yes

Reviewer #2: No

4. Is the manuscript presented in an intelligible fashion and written in standard English?

Reviewer #1: Yes

Reviewer #2: Yes

5. Review Comments to the Author

Reviewer #1: I read with great interest the manuscript title: Non-allergic eye rubbing is a major behavioral risk factor for keratoconus, here authors, confirm what other studies have already shown, that rubbing is a very important cause in the etiopathogenesis of KC.

The authors demonstrate that Eye-rubbing causes CE damage and triggers cellular stress which through its influence on cell apoptosis, migration, and adhesion affects the KTCN phenotype.

Moreover, as in others studies male seems to be more frequents in KC population

I would like to congratulate authors for their work since I really believe Ophthalmologist should stil work hard to inform KC patients of their pathology and risk factors. (Baenninger et al. Do patients with keratoconus have minimal diseade knowledge. Cornea)

Some comments:

-I am not sure that conductive keratoplasty is an therapeutic option nowadays

-Describe acronyms only once in the manuscript

-Please add in results a comparation between male vs female regarding stage of disease and compare in discussion with other papers in the literature

-Compare in more details your results with Moran et al study, especially and with the metaanalysis of Hashemi et al. Cornea 2020

- I would highlight the importance of early diagnosis, especially in children due to the risk of presenting in moderate-advanced stages of diagnosis and risk of progression

References:

1-Rocha de Lossada et al. Acta Ophthalmologica 2020

2-Ertan A & Muftuoglu O. Keratoconus clinical findings according to different age and gender groups. Cornea 2008

3-Leoni-MesplieS, et al. Scalability and severity of keratoconus in children. Am J Ophthalmol 2012

4- Fink BA et al. Differences in Keratoconus as a Function of Gender. Am J Ophthalmol 2005

5-Tharini B, et al. Keratoconus in pre-teen children: Demographics and clinical profile. Indian J Ophthalmol. 2022

Reviewer #2: The manuscript is well written and has some novel findings.

Presence of KC in lower educated population is due to selection bias. Advanced KC prevents people from higher education.

Abstract: "we characterized them as features determining KTCN phenotype, and in particular the corneal epithelium" [lease change the sentence to be more clear especially about epithelium.

6. PLOS authors have the option to publish the peer review history of their article (what does this mean?). If published, this will include your full peer review and any attached files.

Reviewer #1: No

Reviewer #2: **Yes: **Hesam Hashemian

---

## [Author Response · Author response to Decision Letter 0]

15 Feb 2023

Journal Requirements:

Completed as requested.

Completed as requested, the final version of Funding Information and Financial Disclosure is as follows:

‘Supported by National Science Centre in Poland (https://www.ncn.gov.pl/en), Grants 2018/31/B/NZ5/03280 (to MG) and 2021/41/B/NZ5/02245 (to MG), and Polish Ministry of Science and Higher Education, Grant 6817/IA/SP/2018 (to RP and MR; Genomics Core Facility CeNT UW).’

"Supported by National Science Centre in Poland (https://www.ncn.gov.pl/en), Grants 2018/31/B/NZ5/03280 (to MG) and 2021/41/B/NZ5/02245 (to MG)."

Completed as requested, the final version of Role of Funders statement is as follows:

‘The funders had no role in study design, data collection and analysis, decision to publish, or preparation of the manuscript.’

"Next-generation sequencing was performed thanks to Genomics Core Facility CeNT UW using NovaSeq 6000 platform financed by the Polish Ministry of Science and Higher Education (decision no. 6817/IA/SP/2018 of 2018-04-10)."

"Supported by National Science Centre in Poland (https://www.ncn.gov.pl/en), Grants 2018/31/B/NZ5/03280 (to MG) and 2021/41/B/NZ5/02245 (to MG)."

Completed as requested.

Completed as requested. We uploaded and already publish our study’s minimal underlying data set in a public repository accepted by PLOS ONE. Any potentially identifying patient information in this dataset is fully anonymized.

The revised Data Availability statement is as follows:

‘All proceeded clinical/environmental/behavioral/socioeconomic/molecular data, as minimal data set underlying the results described in our manuscript, are shared in a public repository Zenodo (DOI: 10.5281/zenodo.7635600; https://doi.org/10.5281/zenodo.7635600).’

Completed as requested. We uploaded and already publish our study’s minimal underlying data set in public repository accepted by PLOS ONE. Any potentially identifying patient information in this dataset is fully anonymized.

Revised Data Availability statement is as follows:

‘All proceeded clinical/environmental/behavioral/socioeconomic/molecular data, as minimal data set underlying the results described in our manuscript, are shared in a public repository Zenodo (DOI: 10.5281/zenodo.7635600; https://doi.org/10.5281/zenodo.7635600).’

7. We note that Figure 1 includes an image of a participant in the study. 

As per the PLOS ONE policy (http://journals.plos.org/plosone/s/submission-guidelines#loc-human-subjects-research) on papers that include identifying, or potentially identifying, information, the individual(s) or parent(s)/guardian(s) must be informed of the terms of the PLOS open-access (CC-BY) license and provide specific permission for publication of these details under the terms of this license. Please download the Consent Form for Publication in a PLOS Journal (http://journals.plos.org/plosone/s/file?id=8ce6/plos-consent-form-english.pdf). The signed consent form should not be submitted with the manuscript, but should be securely filed in the individual's case notes. 

Please amend the methods section and ethics statement of the manuscript to explicitly state that the patient/participant has provided consent for publication: “The individual in this manuscript has given written informed consent (as outlined in PLOS consent form) to publish these case details”. 

Completed as requested. The signed Consent Form for Publication in a PLOS Journal is securely filed in our individual's case notes. The methods section and ethics statement of the manuscript were revised as follows:

‘The individual whose photographs are presented in Fig 1 in this manuscript has given written informed consent (as outlined in PLOS consent form) to publish the case details.’

8. Please upload a new copy of Figure 3 to 4 and Supporting Figure S1 as the detail is not clear. Please follow the link for more information:

https://blogs.plos.org/plos/2019/06/looking-good-tips-for-creating-your-plos-figures-graphics/

https://blogs.plos.org/plos/2019/06/looking-good-tips-for-creating-your-plos-figures-graphics/

Completed as requested. The figure files were uploaded to the Preflight Analysis and Conversion Engine (PACE) digital diagnostic tool, to ensure that PLOS requirements are met.

Additional Editor Comments:

Both reviewers have indicated that the paper has potential after some corrections. Please read their comments below and address their suggestions.

Reviewers' comments:

Reviewer's Responses to Questions

Comments to the Author

1. Is the manuscript technically sound, and do the data support the conclusions?

Reviewer #1: Yes

Reviewer #2: Yes

2. Has the statistical analysis been performed appropriately and rigorously? 

Reviewer #1: Yes

Reviewer #2: Yes

3. Have the authors made all data underlying the findings in their manuscript fully available?

Reviewer #1: Yes

Reviewer #2: No

We revised that aspect to make our data fully available. As the all proceeded data, constituting our study’s minimal underlying data set is extensive, we uploaded and already published the data in a public repository accepted by PLOS ONE. Any potentially identifying patient information in this dataset is fully anonymized.

The revised Data Availability statement is as follows:

‘All proceeded clinical/environmental/behavioral/socioeconomic/molecular data, as minimal data set underlying the results described in our manuscript, are shared in a public repository Zenodo (DOI: 10.5281/zenodo.7635600; https://doi.org/10.5281/zenodo.7635600).’

4. Is the manuscript presented in an intelligible fashion and written in standard English?

Reviewer #1: Yes

Reviewer #2: Yes

5. Review Comments to the Author

Reviewer #1: I read with great interest the manuscript title: Non-allergic eye rubbing is a major behavioral risk factor for keratoconus, here authors, confirm what other studies have already shown, that rubbing is a very important cause in the etiopathogenesis of KC.

The authors demonstrate that Eye-rubbing causes CE damage and triggers cellular stress which through its influence on cell apoptosis, migration, and adhesion affects the KTCN phenotype.

Moreover, as in others studies male seems to be more frequents in KC population

I would like to congratulate authors for their work since I really believe Ophthalmologist should stil work hard to inform KC patients of their pathology and risk factors. (Baenninger et al. Do patients with keratoconus have minimal diseade knowledge. Cornea)

Some comments:

-I am not sure that conductive keratoplasty is an therapeutic option nowadays

We thank the Reviewer for this valuable comment. We change the original sentence as follows:

“Penetrating keratoplasty (PK), deep anterior lamellar keratoplasty (DALK), and Bowman layer transplantation (BL transplantation) are treatment options in advanced stages of the disease[5]”. 

-Describe acronyms only once in the manuscript

Completed as requested.

-Please add in results a comparation between male vs female regarding stage of disease and compare in discussion with other papers in the literature

Completed as requested. The comparison between male and female patients with KTCN regarding stage of disease, including clinical parameters such as K1, K2, Kmax, anterior and posterior elevations, and TCT are presented in the S1 Table. No statistical differences between genders were identified. 

Results of comparison between male and female patients with KTCN are attached below. 

 Males with KTCN (n=97) Females with KTCN (n=21)

 x ± SD Median x± SD Median

UDVA OD 0.49 ± 0.38 0.40 0.32 ± 0.28 0.20

BDVA OD 0.81 ± 0.35 0.90 0.73 ± 0.34 0.85

UDVA OS 0.43 ± 0.34 0.30 0.52 ± 0.37 0.40

BDVA OS 0.77 ± 0.35 0.90 0.81 ± 0.28 0.90

K1 OD [D] 45.18 ± 4.87 43.90 46.13 ± 6.31 44.0

K2 OD [D] 47.85 ± 5.99 46.10 49.07 ± 6.73 47.20

Kmax OD [D] 53.68 ± 9.32 51.30 55.02 ± 10.61 51.40

anterior elevation OD [μm] 23.14 ± 18.92 17.00 25.43 ± 16.27 24.00

posterior elevation OD [μm] 49.17 ± 36.42 39.00 58.05 ± 44.50 61.00

TCT OD [µm] 466.39 ± 54.28 473.50 460.62 ± 64.50 485.00

thinnest epithelial thickness OD [μm] 43.09 ± 5.45 44.00 43.50 ± 5.55 44.00

K1 OS [D] 45.54 ± 4.94 43.65 44.96 ± 3.43 44.40

K2 OS [D] 48.41 ± 5.91 46.45 47.84 ± 4.16 46.50

Kmax OS [D] 54.76 ± 9.49 52.15 53.51 ± 7.56 51.50

anterior elevation OS [μm] 26.44 ± 19.02 23.50 24.55 ± 15.42 20.50

posterior elevation OS [μm] 52.82 ± 33.64 50.00 52.25 ± 27.45 46.50

TCT OS [µm] 469.76 ± 54.66 473.50 474.80 ± 49.10 485.00

thinnest epithelial thickness OS [μm] 43.28 ± 5.44 44.00 44.57 ± 6.68 46.00

AL OD [mm] 24.03 ± 0.88 24.00 23.86 ± 0.91 23.60

AL OS [mm] 23.96 ± 0.85 23.90 23.76 ± 1.03 23.55

IOP OD [mmHg] 13.26 ± 3.16 13.00 12.72 ± 3.30 12.00

IOP OS [mmHg] 13.16 ± 3.30 13.00 12.94 ± 3.35 12.50

Abbreviations and symbols in the Table: x – average, SD – standard deviation, OD – oculus dexter, OS - oculus sinister, UDVA – uncorrected distance visual acuity, BDVA – best-corrected distance visual acuity, K1 – flat keratometric readings, K2 – steep keratometric readings, Kmax – maximum simulated keratometry, TCT – thinnest corneal thickness, AL - Axial length, IOP – intraocular pressure

We additionally added the following sentences to the Results in the revised manuscript:

‘No statistical differences in clinical parameters such as K1, K2, Kmax, anterior and posterior elevations, and TCT, have been found in comparison between male and female patients with KTCN, and between adult and adolescent patients with KTCN (S1 and S2 Tables).’

We also performed additional patients’ subgroup comparison analyses (males vs. females, adults vs. adolescents), and the findings are presented in S1, S2, S4 and S5 Tables, and we added the following statements: 

‘No statistical differences in clinical parameters such as K1, K2, Kmax, anterior and posterior elevations, and TCT, have been found in comparison between male and female patients with KTCN, and between adult and adolescent patients with KTCN (S1 and S2 Tables).’

‘All analysed data are compiled in S3 Table, while additional results of comparison of male and female patients with KTCN and adult and adolescent patients with KTCN in aspect of selected behavioral, environmental, and socioeconomic factors are presented in S4 and S5 Tables, respectively.’

‘While we did not found differences in the manner of eye rubbing between adult and adolescent patients with KTCN, there was a significant difference considering sex of patients, that is male patients more frequently used knuckles, a base of hand, all fingers and/or a fist (as presented in photographs no. 5-8) (S4 and S5 Tables).’

‘Regarding the working environment, despite the lack of difference in the type of professional occupation between studied groups, we found that the presence of dust in the working environment was more frequently indicated by patients with KTCN (Table 2), and especially male patients with KTCN (S4 Table).’

-Compare in more details your results with Moran et al study, especially and with the metaanalysis of Hashemi et al. Cornea 2020

On our list of references the work of Moran et al. 2020 is numbered as [30]. This reference is already brought to discussion considering the following risk factors: eye rubbing, allergy and sleep position:

‘Previously the strong relation between eye rubbing and KTCN was reported in the case reports[32], case series[33], multivariate analyses[30], and a meta-analysis[9]. Here we verified this aspect and further characterized the eye rubbing behavior practiced by patients with KTCN.’

‘Here, the allergy has not been disclosed as a risk factor for KTCN. Regarding the aspect of allergy and atopy in patients with KTCN, it has been previously discussed as more prevalent in KTCN than in general population[30,44,45].’

‘Other, previously discussed factors as sleeping position and creating additional pressure on the eyeball[30] or smoking[31] weren’t confirmed in our study.’

The meta-analysis of Hashemi et al. 2020 is numbered on our reference list as [9].

We revised the statement in the Discussion as follows:

‘Previously the strong relation between eye rubbing and KTCN was reported in the case reports[32], case series[33], multivariate analyses[30], and a meta-analysis[9].

- I would highlight the importance of early diagnosis, especially in children due to the risk of presenting in moderate-advanced stages of diagnosis and risk of progression

We thank the Reviewer for this valuable comment. To address aspect of importance of early diagnosis we revised the following sentences in the Discussion:

‘We did not identify clinical features corelated with gender or age, however, the gender and age could potentially influence study results regarding the life habits.’ 

and we added the statement:

‘The absence of differences in clinical parameters between adult and adolescent patients indicates a rapid progression in the latter, pointing to the importance of increasing patient awareness as well as early diagnosis due to substantial risk of progression.’

References:

1-Rocha de Lossada et al. Acta Ophthalmologica 2020

2-Ertan A & Muftuoglu O. Keratoconus clinical findings according to different age and gender groups. Cornea 2008

3-Leoni-MesplieS, et al. Scalability and severity of keratoconus in children. Am J Ophthalmol 2012

4- Fink BA et al. Differences in Keratoconus as a Function of Gender. Am J Ophthalmol 2005

5-Tharini B, et al. Keratoconus in pre-teen children: Demographics and clinical profile. Indian J Ophthalmol. 2022

We thank the Reviewer for pointing to the publications, which have strengthened the quality of our discussion.

We added additional sentences to the discussion:

‘We did not identify clinical features corelated with sex or age, however, the sex and age could potentially influence study results regarding the life habits. The absence of differences in clinical parameters between adult and adolescent patients indicates a rapid progression in the latter[57–59], pointing to the importance of increasing patient awareness[60] as well as early diagnosis due to substantial risk of progression.’

‘Moreover, the absence of differences in clinical parameters between adult and adolescent patients, and male and female patients, in contrast to previous reports[38,61], enabled a superior appreciation of both, behavioral and molecular findings.’

Other incorporated changes to the manuscript include: 1) the order and number of the references (seven additional references); 2) the number and order of the supplementary materials in the Supporting Information (four additional tables); 3) the number and order of figures (one additional figure cut out from the original Table 5); and 4) minor grammatical changes, as highlighted in the revised version.

Reviewer #2: The manuscript is well written and has some novel findings.

Presence of KC in lower educated population is due to selection bias. Advanced KC prevents people from higher education.

We thank the Reviewer for this valuable comment. The following sentence was added to the Discussion:

‘In this paper, we presented the level of higher education as the only variable reducing the risk of KTCN (OR=0.52, univariate analysis), but this result should be treated with caution as it may be a consequence of selection bias.’

Abstract: "we characterized them as features determining KTCN phenotype, and in particular the corneal epithelium" [lease change the sentence to be more clear especially about epithelium.

Completed as requested.

The original text:

“Since the environmental, behavioral, and socioeconomic factors in the etiology of keratoconus (KTCN) remain poorly understood, we characterized them as features determining KTCN phenotype, and in particular the corneal epithelium (CE).” 

We change to:

“Since the environmental, behavioral, and socioeconomic factors in the etiology of keratoconus (KTCN) remain poorly understood, we characterized them as features influencing KTCN phenotype, and especially affecting the corneal epithelium (CE).” 

We also performed additional patients’ subgroup comparison analyses (males vs. females, adults vs. adolescents), and the findings are presented in S1, S2, S4 and S5 Tables, and we added the following statements: 

‘No statistical differences in clinical parameters such as K1, K2, Kmax, anterior and posterior elevations, and TCT, have been found in comparison between male and female patients with KTCN, and between adult and adolescent patients with KTCN (S1 and S2 Tables).’

‘All analysed data are compiled in S3 Table, while additional results of comparison of male and female patients with KTCN and adult and adolescent patients with KTCN in aspect of selected behavioral, environmental, and socioeconomic factors are presented in S4 and S5 Tables, respectively.’

‘While we did not found differences in the manner of eye rubbing between adult and adolescent patients with KTCN, there was a significant difference considering sex of patients, that is male patients more frequently used knuckles, a base of hand, all fingers and/or a fist (as presented in photographs no. 5-8) (S4 and S5 Tables).’

‘Regarding the working environment, despite the lack of difference in the type of professional occupation between studied groups, we found that the presence of dust in the working environment was more frequently indicated by patients with KTCN (Table 2), and especially male patients with KTCN (S4 Table).’

Other incorporated changes to the manuscript include: 1) the order and number of the references (seven additional references); 2) the number and order of the supplementary materials in the Supporting Information (four additional tables); 3) the number and order of figures (one additional figure cut out from the original Table 5); and 4) minor grammatical changes, as highlighted in the revised version.

6. PLOS authors have the option to publish the peer review history of their article (what does this mean?). If published, this will include your full peer review and any attached files.

Do you want your identity to be public for this peer review? For information about this choice, including consent withdrawal, please see our Privacy Policy.

Reviewer #1: No

Reviewer #2: Yes: Hesam Hashemian

---

## [Decision Letter · Decision Letter 1]

3 Apr 2023

Non-allergic eye rubbing is a major behavioral risk factor for keratoconus

PONE-D-22-34939R1

Dear Dr. Gajecka,

We’re pleased to inform you that your manuscript has been judged scientifically suitable for publication and will be formally accepted for publication once it meets all outstanding technical requirements.

Kind regards,

Georgios Labiris, MD, PhD

Academic Editor

PLOS ONE

Reviewers' comments:

Reviewer's Responses to Questions

**Comments to the Author**

1. If the authors have adequately addressed your comments raised in a previous round of review and you feel that this manuscript is now acceptable for publication, you may indicate that here to bypass the “Comments to the Author” section, enter your conflict of interest statement in the “Confidential to Editor” section, and submit your "Accept" recommendation.

Reviewer #1: All comments have been addressed

Reviewer #2: All comments have been addressed

2. Is the manuscript technically sound, and do the data support the conclusions?

Reviewer #1: Yes

Reviewer #2: Yes

3. Has the statistical analysis been performed appropriately and rigorously? 

Reviewer #1: Yes

Reviewer #2: I Don't Know

4. Have the authors made all data underlying the findings in their manuscript fully available?

Reviewer #1: Yes

Reviewer #2: No

5. Is the manuscript presented in an intelligible fashion and written in standard English?

Reviewer #1: Yes

Reviewer #2: Yes

6. Review Comments to the Author

Reviewer #1: The authors have resolved all the comments that had been made. In my opinion the work is much improved. Congratulations again and thank you for such a great effort

Reviewer #2: The reviewers' comments were answered as possible. Although the topic lacks novelty it can be of interest for some readers. The manuscript can be considered for publication.

7. PLOS authors have the option to publish the peer review history of their article (what does this mean?). If published, this will include your full peer review and any attached files.

Reviewer #1: No

Reviewer #2: **Yes: **Hesam Hashemian

---

## [Editor Report · Acceptance letter]

5 Apr 2023

PONE-D-22-34939R1 

Non-allergic eye rubbing is a major behavioral risk factor for keratoconus 

Dear Dr. Gajecka:

I'm pleased to inform you that your manuscript has been deemed suitable for publication in PLOS ONE. Congratulations! Your manuscript is now with our production department. 

Kind regards, 

on behalf of

Dr. Georgios Labiris 

Academic Editor

PLOS ONE